# *EgoVLA*: Learning Vision-Language-Action Models from Egocentric Human Videos

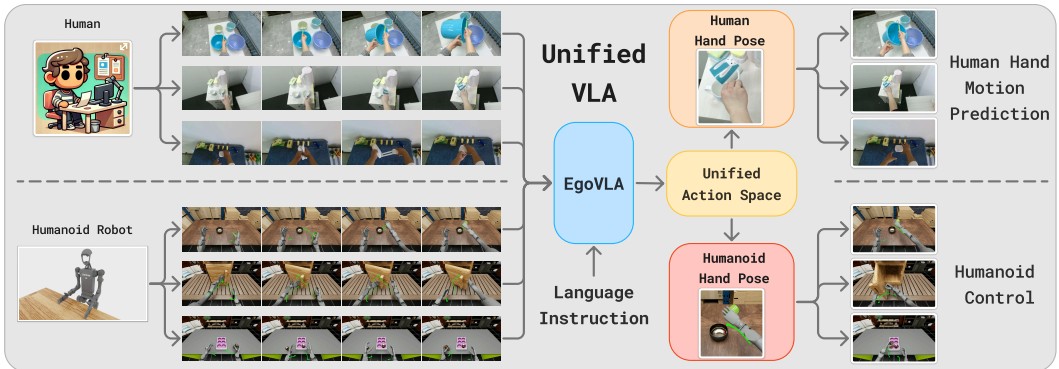

Figure 1: **EgoVLA.** Our vision-language-action model learns manipulation skills from egocentric human videos and transfers them to a bimanual humanoid robot. The top row illustrates the diverse manipulation behaviors demonstrated by humans in the video dataset, while the bottom row shows the robot performing egocentric dexterous manipulation based on the learned skills.

## Abstract

Real robot data collection for imitation learning has led to significant advances in robotic manipulation. However, the requirement for robot hardware in the process fundamentally constrains the scale of the data. In this paper, we explore training Vision-Language-Action (VLA) models using egocentric human videos. The benefit of using human videos is not only for their scale but more importantly for the richness of scenes and tasks. With a VLA trained on human video that predicts human wrist and hand actions, we can perform Inverse Kinematics and retargeting to convert the human actions to robot actions. We fine-tune the model using a few robot manipulation demonstrations to obtain the robot policy, namely *EgoVLA*. We propose a simulation benchmark called *Ego Humanoid Manipulation Benchmark*, where we design diverse bimanual manipulation tasks with demonstrations. We fine-tune and evaluate *EgoVLA* with *Ego Humanoid Manipulation Benchmark* and show significant improvements over baselines and ablate the importance of human data.

## 1 Introduction

There has been a vast advancement in robotic manipulation in the last few years, thanks to large-scale real robot data collection (Vuong et al., 2023; Khazatsky et al., 2024). Compared to approaches that leverage simulation, directly performing supervised learning with real robot data avoids the Sim2Real domain gap and easily increases the task complexity. To efficiently collect complex robot manipulation data, multiple teleoperation tools with joint mapping (Iyer et al., 2024; Dass et al., 2024; Zhao et al., 2023b), exoskeleton (Zhao et al., 2023a; Fang et al., 2024; Yang et al., 2024b), and VR devices (Naceri et al., 2021; Cheng et al., 2024; Ding et al., 2024) are proposed. While this is encouraging, the requirement of a robot and an expert operator fundamentally constrains the scale of the data to be collected.

How about learning manipulation from human videos? If we consider human as a special format of robot, there are 8 billion robots continuously operating across the world in all the environments that

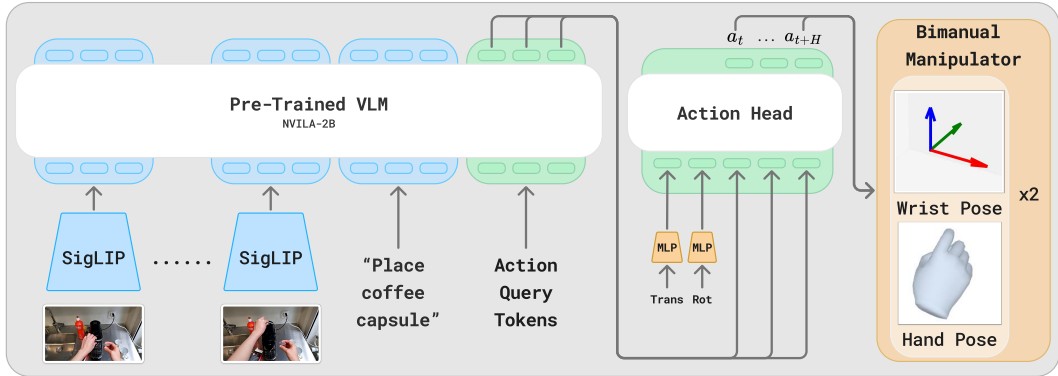

Figure 2: **EgoVLA** takes visual history, language instruction, and action query token as input. The latent features are converted to human action with the action head. We use the wrist pose and MANO hand parameterRomero et al. (2017) as human action space.

we would like robots to operate in. Recent work on Hand-Object Interaction prediction (Tian et al., 2024) has shown promising results in forecasting long-term human intention on manipulation. If we are able to leverage such human data to train a robot policy, it easily scales up not only the number of training data, but more importantly the diversity of tasks and scenes. It allows training on scenes where the current robot might not easily fit in or tasks that are even challenging for teleoperation.

Our key observation is: The difference between human action and robot action spaces might not be that large, and it can be approximated by a few geometric transforms. Instead of training a Robot Vision-Language-Action (VLA) model on robot data (Octo Model Team et al., 2024; Kim et al., 2024; Black et al., 2024; Brohan et al., 2023), we propose to train a Human Egocentric VLA (*EgoVLA*) on human data. Specifically, given a few frames of visual observation language instruction, and current hand pose as input, the VLA will predict human actions in a few steps in the future. The action space includes the human wrist and hand joint angles. This human action space can be converted to a robot action space by converting wrist location to end-effector location through Inverse Kinematics and converting human hand joints to robot hand joints through retargeting. Thus the Human VLA is inherently already a robot policy, except the inputs are human hand images and there is still error in action outputs. We can correct this by further fine-tuning the VLA with a few robot demonstrations collected by teleoperation. In this way, we do not require large-scale robot data for training.

To benchmark robotic manipulation, we propose a new humanoid bimanual manipulation benchmark based on NVIDIA IsaacSim(NVIDIA, 2024a), namely *Ego Humanoid Manipulation Benchmark*. In this benchmark, we provide 12 tasks including simple tasks that perform atomic actions and long-horizon tasks that combine multiple atomic actions. We collect 100 demonstrations per task and evaluate models using the benchmark. In our experiments, we first train our *EgoVLA* on our Ego-Centric Human Manipulation dataset and fine-tune on the humanoid manipulation demonstration collected for specific tasks. In our simulation benchmark, our *EgoVLA* outperforms different specialist and generalist baselines in both short-horizon and long-horizon tasks and achieves better generalization across visual observation and spatial locations.

## 2 RELATED WORK

**Dexterous Manipulation:** Research in dexterous manipulation has progressed from control-based methods (Rodriguez et al., 2012; Rosales et al., 2012; Prattichizzo et al., 2012; Ponce et al., 1993; 1997; Zheng & Chew, 2009; Dai et al., 2018) to learning-driven approaches (Andrychowicz et al., 2020; Nagabandi et al., 2020). While early work emphasized precision, generalization across diverse scenarios remained limited. Learning-based methods introduced pose vector generation (Jiang et al., 2021; Corona et al., 2020; Yang et al., 2021), intermediate representations (Shao et al., 2020; Wu et al., 2022), and contact maps (Brahmbhatt et al., 2019; Turpin et al., 2022), but large-scale dexterous manipulation remains an open challenge. Recent efforts leverage egocentric human videos for

training task-specific policies (Qiu et al., 2025; Kareer et al., 2024). In contrast, we aim to develop generalist manipulation models directly from egocentric human demonstrations.

**Vision-Language-Action (VLA):** Vision-Language Models (VLMs)(Achiam et al., 2023; Lin et al., 2024; Tong et al., 2024) have shown strong generalization across multimodal tasks(Pratt et al., 2023; Alaluf et al., 2025; Kuo et al., 2023; Huang et al., 2025; Lv et al., 2023). Building on this, Vision-Language-Action Models (VLAs)(Brohan et al., 2023; Kim et al., 2024; Wen et al., 2024; Octo Model Team et al., 2024; Black et al., 2024; Intelligence et al., 2025) fine-tune VLMs with large-scale robot data, enabling perception-action integration. However, VLA training is data-intensive, often requiring extensive teleoperation(Mandlekar et al., 2018; 2023) or scripted execution (Dasari et al., 2019; Kalashnikov et al., 2018). OpenVLA (Kim et al., 2024) and Octo (Octo Model Team et al., 2024) leverage crowd-sourced robot datasets (Vuong et al., 2023) but face scalability bottlenecks. We propose an alternative: learning policies from human egocentric videos, augmented by small-scale target-domain fine-tuning.

**Egocentric Vision:** Egocentric vision research (Damen et al., 2022; Sigurdsson et al., 2018; Li et al., 2015) traditionally faced limitations in data scale and diversity. Recent datasets (Grauman et al., 2022; 2024) improve coverage but focus on activities beyond current robotic capabilities. Simpler datasets (Mahdisoltani et al., 2018; Damen et al., 2018) capture everyday interactions but lack pose annotations. We address this by curating a targeted mixture of datasets and introducing an Egocentric Human Video Dataset (Section 3.1) optimized for dexterous manipulation learning.

**Learning from In-the-Wild Video:** Several works (Bahl et al., 2023; Wen et al., 2023) propose extracting affordances or interaction cues from in-the-wild videos. Motivated by egocentric vision, recent studies (Nair et al., 2022; Majumdar et al., 2023; Karamcheti et al., 2023; Yang et al., 2024a; Zeng et al., 2024; Ye et al., 2024; Lirui et al., 2024) pretrain representations using human videos, showing positive transfer. However, most focus on unsupervised learning without leveraging fine-grained hand or wrist pose information. Our work instead uses high-quality egocentric data within the VLA framework to directly improve dexterous policy learning, capitalizing on advances in wearable hand-tracking technologies.

## 3 LEARNING MANIPULATION SKILLS FROM EGO-CENTRIC HUMAN VIDEOS

In this section, we describe the construction of our egocentric human manipulation dataset, the training of *EgoVLA* on this dataset, bridging the embodiment gap between humans and humanoid robots, and the deployment of *EgoVLA* for manipulation tasks.

### 3.1 EGO-CENTRIC HUMAN MANIPULATION DATASET

Following insights from language model and vision-language model training, we emphasize the importance of dataset structure in driving model performance. We construct a large-scale human egocentric manipulation dataset focused on skill-rich video sequences with accompanying pose annotations. The combined dataset contains egocentric RGB observations, wrist poses, hand poses, and camera poses. Our dataset integrates sequences from four sources with their relative proportions shown in Fig. 3: **HOI4D** consists of 4,000 videos capturing one-handed manipulations such as pick-and-place, reorientation, and articulated object interaction. **HOT3D** provides 833 minutes of video interacting with 33 rigid objects, with precise 3D hand and camera pose annotations. **HoloAssist** offers 166 hours of recordings of complex tasks such as battery replacement, furniture assembly, and machine setup. Although its hand pose annotations are noisier, it captures rich bimanual interactions. To avoid overrepresentation of HoloAssist, which has noisier labels, we uniformly sampled 1/10 of it to balance tasks and sources. **TACO** includes 2,317 motion sequences covering 151 tool-action-object triplets.

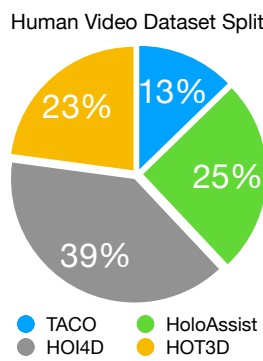

Human Video Dataset Split

Figure 3: **Human Data**

**Data Processing:** Egocentric videos introduce challenges for learning due to continuous camera motion. To mitigate this, we use world-frame camera poses to project future wrist positions into the current camera frame, ensuring consistent supervision. Training samples are generated by sampling

RGB observations at 3 FPS, balancing computational efficiency and temporal continuity. In total, our dataset comprises approximately 500,000 image-action pairs across diverse manipulation tasks.

**Language Label:** The combined dataset includes ego-centric RGB visual observations, wrist poses, hand poses, and camera poses. Notably, HoloAssist, TACO and HOI4D provide specific language labels for each video clip, while HOT3D lacks language annotations. To maintain consistency across datasets, we include placeholder language instructions for HOT3D.

**Hand Pose Annotations:** For the human datasets used in this work, TACO (Liu et al., 2024), HOT3D (Banerjee et al., 2024), and HOI4D (Liu et al., 2022b) provide MANO (Romero et al., 2017) annotations directly, while for HoloAssist (Wang et al., 2023), which employs a different hand model, we retarget the provided poses to the MANO representation to ensure a unified parameterization. To further streamline the problem and focus on learning manipulation skills rather than hand appearance, we use the average human hand shape provided by MANO.

## 3.2 *EgoVLA* MODEL

We build *EgoVLA* on top of a vision-language model to leverage strong visual and semantic reasoning. Specifically, we use *NVILA-2B*(Liu et al., 2025) as the backbone for its robust vision-language understanding and compact size, enabling both intention inference and efficient fine-tuning. As shown in Fig.2, *EgoVLA* takes as input current and historical egocentric visual observations, language instructions, action query tokens, and human proprioception. These inputs are encoded by the VLM backbone and further processed by an action head to predict future human or robot actions.

Visual Observations consists of six RGB frames: the current observation and five preceding frames, sampled at $0.2$sec intervals, covering a $1$sec history. Each frame has a resolution of $384 \times 384$. Language Instructions describe the immediate desired behavior. This design focuses the model on skill execution rather than high-level planning, ensuring a clear mapping between language input and predicted actions. Human Proprioception State includes wrist translations/rotations, and hand pose parameters. These are processed through MLPs before being passed to the action head.

Each predicted action includes the wrist pose (3D translation and rotation in rot6D representation (Zhou et al., 2020) in the camera frame) and hand joint angles, represented using the top 15 PCA components of the MANO hand model (Romero et al., 2017).

**Training objective:** *EgoVLA* is trained to regress future wrist poses in the camera frame and hand joint parameters. The full training objective is defined as follows:

$$\mathcal{L} = \lambda_{\text{wrist trans}}\mathcal{L}_{\text{wrist trans}} + \lambda_{\text{wrist rot}}\mathcal{L}_{\text{wrist rot}} + \lambda_{\text{joint}}\mathcal{L}_{\text{joint}}$$

where $\mathcal{L}_{\text{wrist trans}} = \|\mathbf{T}_{\text{pred}} - \mathbf{T}_{\text{gt}}\|_2^2$ is L2 loss for wrist translation regression. $\mathbf{T}_{\text{pred}}$ is the predicted wrist translation in the camera frame, and $\mathbf{T}_{\text{gt}}$ is the ground-truth wrist translation. $\mathcal{L}_{\text{wrist rot}} = \|\mathbf{R}_{\text{pred}} - \mathbf{R}_{\text{gt}}\|_2^2$ is the rotation loss for wrist rotation prediction where $\mathbf{R}_{\text{pred}}$ is the predicted wrist rotation in the camera frame, and $\mathbf{R}_{\text{gt}}$ is the ground-truth wrist rotation. Our model predicts rotation in rot6d(Zhou et al., 2020), and we convert the predicted rotation to rotation matrix before compute the loss. $\mathcal{L}_{\text{joint}} = \|\mathbf{\Theta}_{\text{pred}} - \mathbf{\Theta}_{\text{gt}}\|_2^2$ is L2 loss for hand joint angle regression, where $\mathbf{\Theta}_{\text{pred}}$ are the predicted MANO parameters, and $\mathbf{\Theta}_{\text{gt}}$ are the ground-truth MANO parameters. $\lambda_{\text{joint}}, \lambda_{\text{wrist trans}}, \lambda_{\text{wrist rot}}$ are weights to balance different terms.

**Action Head & Action Query Tokens:** The action head is a transformer (300M) consisting of six encoder layers, each with a hidden size of 1536. It takes as input the human (or robot) proprioception state and the latent embeddings corresponding to the action query tokens, and predicts a sequence $A_t = [a_t, a_{t+1}, \ldots, a_{t+H}]$ over a 1-second horizon (30 future steps at 30 Hz) for both hands. We use the last $H = 30$ word IDs from the vocabulary as action query tokens.

**Training Details:** We first pretrain *EgoVLA* on our Ego-Centric Human Manipulation Dataset for 20 epochs. This is followed by 115 epochs of post-training on robot demonstration data, where the learning rate is reduced after 100 epochs. During training, the full model, including the visual encoder, is fine-tuned.Further training configurations are detailed in the Supplementary Materials.

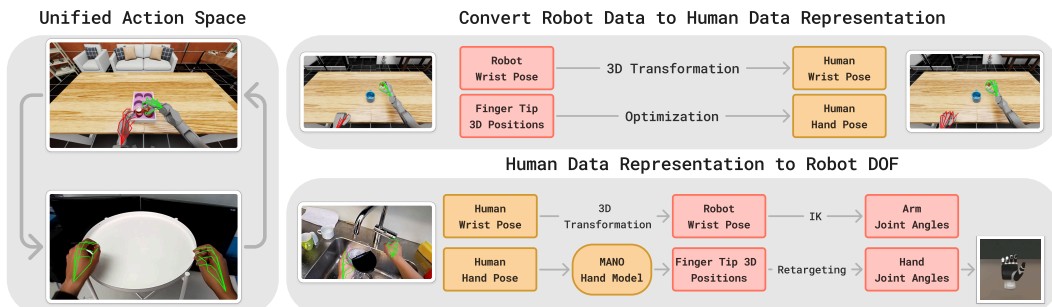

Figure 4: **Unified Action Space:** MANO hand parameters are used as a shared action space for humans and robots. For robot hands, during training, optimized mano parameters produce the same fingertip position as the robot hand fingertip. A small MLP maps predicted finger tip positions to joint commands during deployment.

### 3.3 TRANSFERRING *EgoVLA* TO HUMANOID ROBOT

Humans and humanoid robots share a similar manipulation framework using two arms and hands. However, directly transferring *EgoVLA* to humanoid robots is challenging due to differences in camera pose, hand morphology, and visual appearance. To enable the deployment, we fine-tune *EgoVLA* using a small set of robot demonstrations, leveraging a unified action space as in Fig. 4 and we provide more details regarding the Humanoid Robot data in Sec. 4

**Retargeting robot data to human representation:** To fine-tune on robot data, we first align the robot's action space with the human representation. For end-effector poses, 3D transformations align the robot and human coordinate systems. Aligning hand configurations is more complex: we estimate MANO (Romero et al., 2017) parameters that best approximate the robot hand's actuation by minimizing the discrepancy between predicted and observed fingertip positions: $\underset{\boldsymbol{\Theta}}{\text{minimize}} \quad \mathcal{L}(\boldsymbol{\Theta}) = \frac{1}{5}\sum_{i=1}^{5} \text{SmoothL1}(\mathbf{J}_{\text{pred}}(\boldsymbol{\Theta})_i, \mathbf{J}_{\text{obs},i})$ where $\boldsymbol{\Theta} \in \mathbb{R}^{15}$ are the MANO hand parameters, $\mathbf{J}_{\text{pred}}(\boldsymbol{\Theta})$ are fingertip positions computed via MANO forward kinematics, and $\mathbf{J}_{\text{obs}} \in \mathbb{R}^{5\times3}$ are the observed robot fingertip positions. This unified action space enables direct fine-tuning of *EgoVLA* on robot demonstrations without requiring additional architectural changes or reinitialization.

**Mapping human hand to robot hand:** At inference time, the wrist and hand poses predicted by *EgoVLA* are mapped to the robot's actuators, as illustrated in Fig. 4 (bottom row). Wrist poses are first converted into robot end-effector poses through 3D transformations, and the corresponding arm joint angles are solved via inverse kinematics (IK). For hand actuation, we use the MANO model to calculate 3D hand keypoints from the predicted MANO parameters. A lightweight MLP then predicts the robot's hand joint commands given 3D hand keypoints. This MLP is trained on robot demonstrations where hand actuations are retargeted into human hand representations. This mapping achieves a mean fingertip position error of $5 \times 10^{-5}, \text{m}$. Furthermore, replaying raw demonstrations through this retargeting pipeline preserves task validity, indicating that the small errors introduced during retargeting do not significantly affect control performance. Additional implementation details are provided in the Supplementary Materials.

## 4 *Ego Humanoid Manipulation Benchmark*

Beyond data scarcity, a major challenge in learning-based robotics is the lack of scalable, robust, and reproducible evaluation. Real-world evaluation is often costly, time-consuming, and raises concerns around safety and reproducibility—barriers that disproportionately affect resource-constrained settings such as academic labs. Recent work (Li et al., 2024) has shown that simulation-based evaluations are highly correlated with real-world performance, supporting their use as a reliable proxy. To enable consistent benchmarking of humanoid manipulation, we introduce the *Ego Humanoid Manipulation Benchmark*, built using NVIDIA Isaac Lab (Mittal et al., 2023). *Ego Humanoid Manipulation Benchmark* is not designed for direct sim-to-real transfer. Instead, we leverage simulation, similar to LIBERO (Liu et al., 2023) and SIMPLER (Li et al., 2024)—as a controlled and

Figure 5: **Task Visualization.** All simulated tasks with predicted wrist trajs from *EgoVLA*.

reproducible testbed for evaluating manipulation policies. Our simulated platform features the Unitree H1 (Robotics, 2024) humanoid robot with two Inspire dexterous hands (Robots, 2024), and includes 12 manipulation tasks spanning both **Short Horizon** atomic actions (Push-Box, Flip-Mug, Pour-Balls, Close-Drawer, Open-Drawer, Open-Laptop, Stack-Can) and **Long-Horizon**, multi-stage skills (Sort-Cans, Insert-Cans, Unload-Cans, Insert-And-Unload-Cans, Stack-Can-Into-Drawer), as shown in Fig. 5.

**Observation & Action Space:** Our benchmark provides robot joint positions, end-effector poses, contact forces, and egocentric RGB-D visual input for observations. While *EgoVLA* uses only egocentric vision, end-effector poses, hand joint actuation, and task descriptions, additional modalities are available for future research. The robot is controlled via end-effector control for arms and PD joint control for hands. Each hand has 12 DoFs (6 active, 6 mimic joints). The final 36-dimensional action space combines arm inverse kinematics with direct hand actuation. Control frequency is 30 Hz. We also provide per-step success indicators and sub-task completion flags for every task. Definitions and success metrics for each sub-task are detailed in the supplementary materials.

**Diverse Visual Background:** Simulation allows full control over visual conditions. We includes 5 room textures (Room 1–5) and 5 table textures (Table 1–5), generating 25 distinct visual background combinations for robust generalization evaluation.

**Demonstrations:** To support imitation learning, we provide expert demonstrations collected via OpenTelevision (Cheng et al., 2024) using the Meta Quest 3. Demonstrations are collected using Room 1, 2, or 3, with Table 1 fixed. For each task, we collect 100 successful demonstrations, with episode lengths ranging from 100 to 500 frames depending on task complexity.

## 5 EXPERIMENTS

### 5.1 HUMAN MANIPULATION MODELING

Before directly evaluating robotic manipulation performance, we first assess how well *EgoVLA* models human hand motion after training on the Human Ego-Centric Manipulation Dataset. Quantitatively, the average future prediction error for the human wrist translation is approximately $8$, cm. When projected onto the 2D image plane, the normalized error is around $0.13$, comparable to state-of-the-art results reported in HOI-forecast (Liu et al., 2022a). Figure 6 shows a prediction result from *EgoVLA* on the evaluation set, with additional trajectory visualizations provided in the Appendix. To further examine whether *EgoVLA* captures environmental context and follows language instructions, we sample examples from the HOI4D evaluation set, modifying the original language prompts while keeping the visual inputs unchanged. As shown in the third column of Fig. 7, when the instruction changes from *Put it in the drawer* to *Take it out of the drawer*, the predicted hand trajectories shift from moving into the drawer to moving toward the cabinet surface. Similarly, in the second column, altering the instruction from *Put something in it* to *Open and close the door* results in trajectories shifting from placing objects inside the safe to interacting with the safe's door. These results demonstrate that *EgoVLA* not only accurately models human motion, but also learns the underlying semantic intent behind the actions.

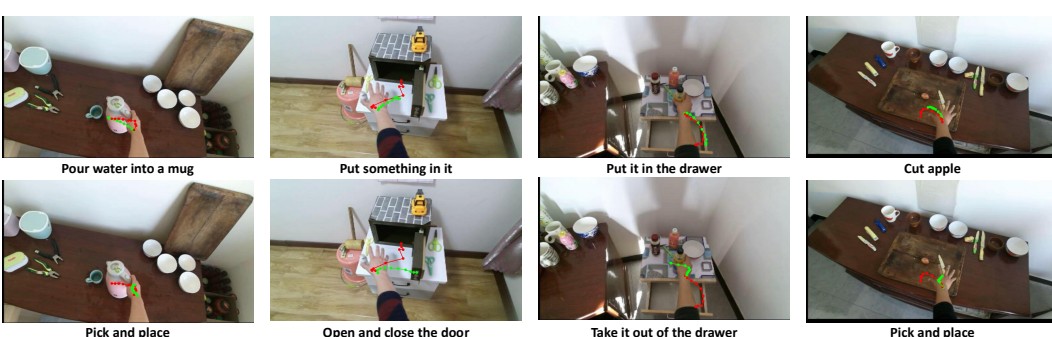

Figure 6: **Human Trajectory Prediction:** The green dotted lines are the future human hand trajectories predicted by our EgoVLA and the red dotted lines are the ground-truth future human hand trajectories.

| Pour water into a mug | Put something in it | Put it in the drawer | Cut apple |
| Pick and place | Open and close the door | Take it out of the drawer | Pick and place |

Figure 7: **Visual Instruction Following.** Top: original HOI4D samples with corresponding language instructions. Red lines indicate ground-truth human wrist trajectories, and green lines denote predictions. Bottom: results with modified instructions while keeping the same visual input. *EgoVLA* adjusts predicted trajectories appropriately according to the new instructions, demonstrating its understanding of language and environment.

Table 1: **Evaluation on Short-Horizon Tasks on Seen/Unseen Visual Configurations**

| Method | Stack-Can SR↑ | PSR↑ | Push-Box SR↑ | PSR↑ | Open-Drawer SR↑ | PSR↑ | Close-Drawer SR↑ | PSR↑ | Flip-Mug SR↑ | PSR↑ | Pour-Balls SR↑ | PSR↑ | Open-Laptop SR↑ | PSR↑ | Mean SR↑ | PSR↑ |
|---|---|---|---|---|---|---|---|---|---|---|---|---|---|---|---|---|
| Seen Visual Configuration | | | | | | | | | | | | | | | | |
| ACT Zhao et al. (2023b) | 22.22 | 22.22 | 11.11 | 85.19 | 18.52 | 66.67 | 48.15 | 96.30 | 7.40 | 7.40 | 3.70 | 77.78 | 62.96 | 62.96 | 24.87 | 59.79 |
| EgoVLA-NoPretrain Lin et al. (2024) | 55.56 | 55.56 | 51.85 | 75.93 | 59.26 | 77.78 | 100.00 | 100.00 | 3.70 | 3.70 | 85.19 | 93.83 | 96.30 | 96.30 | 64.55 | 71.87 |
| *EgoVLA* (50%) | 44.44 | 44.44 | 33.33 | 66.67 | 22.22 | 61.73 | 100.00 | 100.00 | 22.22 | 22.22 | 77.78 | 92.59 | 37.04 | 44.44 | 48.15 | 61.73 |
| ***EgoVLA*** | 77.78 | 77.78 | 70.37 | 83.33 | 59.26 | 81.48 | 100.00 | 100.00 | 59.26 | 59.26 | 77.78 | 92.59 | 100.00 | 100.00 | 77.78 | 84.92 |
| Unseen Visual Configuration | | | | | | | | | | | | | | | | |
| ACT Zhao et al. (2023b) | 9.09 | 10.61 | 18.18 | 87.88 | 24.24 | 71.97 | 63.64 | 96.97 | 0.00 | 0.00 | 6.06 | 59.09 | 53.03 | 53.03 | 24.89 | 54.22 |
| EgoVLA-NoPretrain Lin et al. (2024) | 57.58 | 57.58 | 69.70 | 82.58 | 46.15 | 71.28 | 86.36 | 89.90 | 4.69 | 4.69 | 46.03 | 79.37 | 48.48 | 53.03 | 51.28 | 62.63 |
| ***EgoVLA*** | 62.12 | 62.12 | 75.76 | 84.85 | 50.00 | 75.25 | 98.48 | 98.48 | 30.77 | 30.77 | 83.33 | 94.44 | 83.33 | 87.88 | 69.11 | 76.26 |

## 5.2 HUMANOID ROBOT EVALUATION

We evaluate the manipulation performance of different models using two metrics: *Success Rate (SR)*, measuring overall task success, and *Progress Rate (PSR)*, defined as the average number of completed subtasks relative to the total subtasks in a long-horizon task.

**Baselines:** We compare *EgoVLA* against two baselines: (1) *EgoVLA-NoPretrain*, which fine-tunes the pre-trained VLM on robot demonstrations without human video pretraining; (2) *ACT*, which trains a separate specialist transformer for each task individually. For ACT, we adopt the recommended training settings.

**Evaluation Setup:** We assess model performance under two settings: **Seen**, where visual backgrounds are identical to those encountered during training, and **Unseen**, where backgrounds are entirely novel. To promote robustness, object positions are **randomized** at the start of each rollout, with placement constrained to regions not seen during training. Detailed specifications for background splits and object randomization ranges (up to $20\text{cm} \times 20\text{cm}$) are provided in the supplementary materials. For **Seen** backgrounds, each model is evaluated with 27 rollouts per task: nine episodes for each of three backgrounds. For **Unseen** backgrounds, each model is evaluated with 66 rollouts per task: three episodes for each of 22 unseen backgrounds.

We use simulation to ensure reproducible and robust evaluation, but the method itself is modality-agnostic and can learn effectively from any high-quality teleoperated demonstrations, whether collected in simulation or the real world.

Table 2: **Evaluation on Long-Horizon Tasks on Seen/Unseen Visual Configurations**.

| Method | Insert-And-Unload-Cans | | Stack-Can-Into-Drawer | | Sort-Cans | | Unload-Cans | | Insert-Cans | | Mean | |
| --- | --- | --- | --- | --- | --- | --- | --- | --- | --- | --- | --- | --- |
| | SR ↑ | PSR ↑ | SR ↑ | PSR ↑ | SR ↑ | PSR ↑ | SR ↑ | PSR ↑ | SR ↑ | PSR ↑ | SR ↑ | PSR ↑ |
| Seen Visual Configuration | | | | | | | | | | | | |
| ACT Zhao et al. (2023b) | 0.00 | 11.11 | 11.11 | 37.04 | 0.00 | 28.63 | 0.00 | 35.80 | 0.00 | 19.75 | 2.22 | 26.47 |
| EgoVLA-NoPretrain Lin et al. (2024) | 7.41 | 45.93 | 0.00 | 18.52 | 51.85 | 79.63 | 62.96 | 75.00 | 11.11 | 55.56 | 26.67 | 54.93 |
| *EgoVLA (50%)* | 0.00 | 28.15 | 29.63 | 57.41 | 0.00 | 33.33 | 0.00 | 30.56 | 7.41 | 49.07 | 7.41 | 39.70 |
| ***EgoVLA*** | 44.44 | 77.04 | 40.74 | 75.93 | 55.56 | 88.89 | 66.67 | 83.33 | 22.22 | 78.70 | 45.93 | 80.78 |
| Unseen Visual Configuration | | | | | | | | | | | | |
| ACT Zhao et al. (2023b) | 0.00 | 7.95 | 1.52 | 33.84 | 0.00 | 20.71 | 1.52 | 33.83 | 0.00 | 21.21 | 0.61 | 23.51 |
| EgoVLA-NoPretrain Lin et al. (2024) | | 29.09 | 0.00 | 20.08 | 15.15 | 45.83 | 34.85 | 55.68 | 6.06 | 30.30 | 11.21 | 36.20 |
| ***EgoVLA*** | 31.82 | 76.97 | 28.79 | 60.98 | 18.18 | 68.94 | 50.00 | 75.76 | 15.15 | 62.88 | 28.79 | 69.11 |

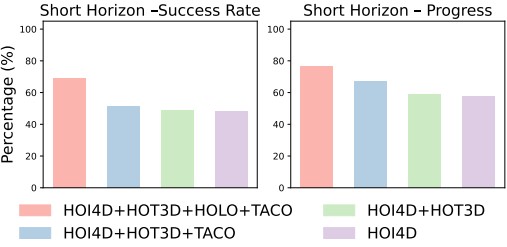

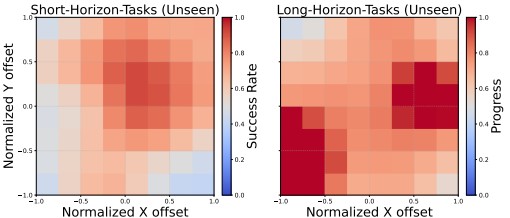

Figure 8: **Data Mixture Ablation.** *EgoVLA* pretrained on different mixtures of human egocentric datasets, evaluated on **Unseen** visual backgrounds for short-horizon tasks. Greater diversity consistently improves generalization performance.

Figure 9: **Spatial Distribution.** Success rate and progress of *EgoVLA* under **Unseen** visual backgrounds, visualized across object spawning positions. The model maintains strong performance across a wide area, with higher success in regions commonly associated with effective bimanual manipulation.

***EgoVLA* Requires Robot Data Post-training:** While *EgoVLA* is pretrained on egocentric human videos using a unified action space, zero-shot deployment on humanoid robots without fine-tuning on robot data results in 0% success across all tasks. This failure stems from appearance, perception, and kinematic mismatches between humans and robots. Even for tasks seen during pretraining (e.g., *Pouring Balls*), execution fails without robot-specific adaptation, highlighting the need for post-training on in-domain robot data.

**Ego-centric Human Pretraining Improves In-Domain Performance:** We first evaluate models on **Seen** visual backgrounds. As shown in Table 1 and Table 2, *EgoVLA* consistently outperforms the *EgoVLA-NoPretrain* baseline across both short- and long-horizon tasks. Gains are especially pronounced on tasks requiring fine-grained manipulation, such as *Stack-Cans*, *Sort-Cans*, *Insert-And-Unload-Cans*, and *Flip-Mugs*. We attribute this to the human pretraining phase, during which *EgoVLA* learns general manipulation skills involving "hands" independent of embodiment (human or robot). In contrast, *EgoVLA-NoPretrain* learns solely from robot demonstrations without such transferable priors. Moreover, the performance gap between *EgoVLA* and *EgoVLA-NoPretrain* is larger for long-horizon tasks, achieving approximately 20% higher success rates. Compared to the specialist *ACT* baselines, generalist models (*EgoVLA* and *EgoVLA-NoPretrain*) perform substantially better on both short- and long-horizon tasks. This is likely because specialist models must simultaneously learn low-level manipulation and long-horizon planning from scratch, whereas generalist models leverage shared low-level skills across tasks.

**Ego-centric Human Pretraining Enhances Out-of-Domain Generalization:** As shown in Table 1, *EgoVLA* maintains strong generalization in short-horizon tasks on **Unseen** visual backgrounds, with only a minor drop in mean success rate, whereas *EgoVLA-NoPretrain* exhibits a substantial 23% decline. For long-horizon tasks, as shown in Table 2, *EgoVLA* achieves around 30% success on unseen backgrounds. Although success rates decline compared to seen environments, the progress rate remains similar, suggesting that failures primarily occur at the final stages of task execution rather than early subgoals. This demonstrates that pretraining on egocentric human videos significantly improves *EgoVLA*'s ability to generalize to novel environments.

**Ablation Study — Robot Data Scale:** To analyze the contributions of human video pretraining and robot demonstration fine-tuning, we train a variant of *EgoVLA* using only 50% of the robot demonstrations, denoted as *EgoVLA (50%)*. As shown in Table 1 and Table 2, while *EgoVLA (50%)* can

Figure 10: **Robot Trajectories (left to right).** In the top row, the robot inserts two cans into the tray and unloads them, and in the bottom row, the robot puts the can over the saucer and closes the drawer.

still complete some tasks, its overall success rate drops substantially, particularly on long-horizon tasks (from 45.93% to 7.41%). These results suggest that, although pretraining on egocentric human videos significantly improves both in-domain performance and generalization, *EgoVLA* still requires a moderate amount of task-specific robot demonstrations to achieve strong performance.

**Ablation Study - Pretrain Data Mixture:** To further assess the impact of pretraining data composition, we pretrain *EgoVLA* using different data mixtures, as shown in Fig. 8. Results show that increasing the scale and diversity of human pretraining data consistently improves downstream performance, with both average Success Rate (SR) and Progress Rate (PSR) increasing across tasks. Notably, despite noisy hand annotations in HoloAssist, missing language labels in HOT3D, and limited visual diversity in TACO, we still observe positive transfer, highlighting the robustness of our approach to dataset imperfections.

**Performance of *EgoVLA* Conditioned on Object Spawning Positions:** Beyond generalization to diverse visual backgrounds, we further analyze model performance conditioned on object spawning positions. As shown in Fig. 9, we visualize the success rate under *Unseen* visual backgrounds for short-horizon tasks and the average progress for long-horizon tasks. For short-horizon tasks, we observe that success rates are higher when objects are placed closer to the center of the randomized region. For long-horizon tasks, two distinct high-probability regions emerge. We hypothesize that this arises because many long-horizon tasks involve bimanual operations, with each peak corresponding to successful interactions using either the left or right hand.

**Trajectory Visualization:** To complement quantitative evaluation, we visualize policy rollouts for long-horizon tasks in Fig. 10. We display the sequence of visual observations alongside the predicted wrist trajectories produced by *EgoVLA*. For clarity, only the future wrist positions are visualized. Despite training with only 100 demonstrations per task, *EgoVLA* successfully executes diverse long-horizon tasks while demonstrating strong spatial and visual generalization. Additional trajectory visualizations are provided in the supplementary materials.

## 6 CONCLUSION

In this work, we present *EgoVLA*, a vision-language-action model trained on egocentric human videos for dexterous manipulation. *EgoVLA* is developed through pretraining a vision-language model on large-scale human egocentric manipulation dataset, and fine-tuning on a small set of robot demonstrations. To enable transfer across embodiments, we introduce a unified action space that aligns human and robot hand representations. Experimental results demonstrate that human video pretraining enables *EgoVLA* to learn a generalist manipulation policy, achieving strong performance across diverse tasks with limited robot data, with strong generalization ability.

## 7 LIMITATION

Our pretraining framework requires human data with hand and wrist pose annotations, which may limit data availability. However, the increasing accessibility of high-fidelity AR/VR devices (e.g., Quest 3, Vision Pro, Aria Glasses) is expected to ease this constraint. Additionally, although *EgoVLA* is pretrained with a unified action space, it cannot be directly deployed for manipulation without further fine-tuning on a moderate amount of robot data. Future work may explore improving zero-shot transferability through more embodiment-agnostic pretraining.

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

# Appendix

## A USE OF LLM

In preparing this manuscript, we made limited use of a large language model (LLM) to assist with grammar correction, sentence rephrasing, and improving the readability of the text. All scientific content, technical ideas, and original writing were produced by the authors. Any text that was polished with the help of the LLM was carefully reviewed and verified by the authors to ensure accuracy and faithfulness to the intended meaning.

## B DATASET DETAILS

Section 3.1 presents an overview of our Ego-Centric Human Video Dataset. Here, we provide a detailed visualization of the task distribution for HOI4D (Liu et al., 2022b) and HoloAssist (Wang et al., 2023). The task distribution for HOT3D(Banerjee et al., 2024) is not visualized, as task labels are not provided for its sequences.

For HOI4D (Liu et al., 2022b), we show the task descriptions across all sequences, while for HoloAssist (Wang et al., 2023) and TACO (Liu et al., 2024), we illustrate the verbs from the action descriptions. To enhance clarity, a logarithmic scale is applied to the frequency distributions for both datasets. As shown in Fig. 11, Fig. 12, and Fig. 13, all datasets provide ego-centric human video across diverse tasks.

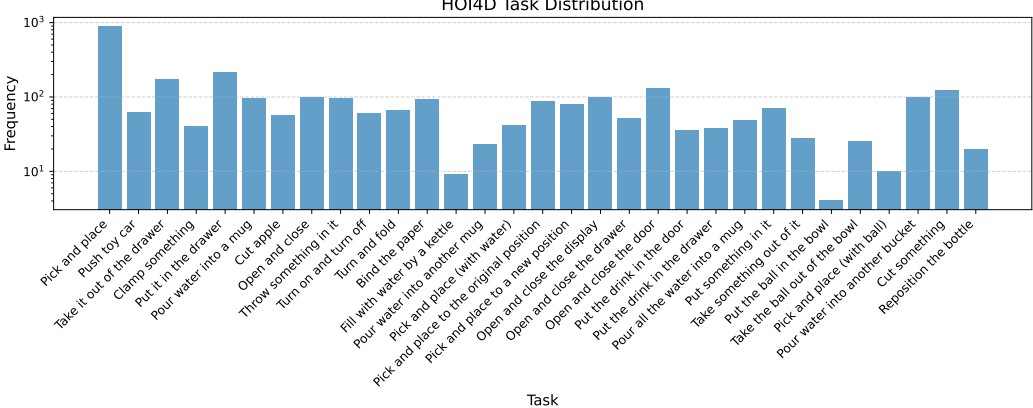

Figure 11: HOI4D (Liu et al., 2022b) Task distribution

## C TRAINING DETAILS

### C.1 *EgoVLA* TRAINING DETAILS

We provide detailed training configurations to ensure the faithful reproduction of our results. All models were trained using 32 A100 GPUs.

We use the following hyperparameters for training on our ego-centric human video dataset and during fine-tuning with robot demonstrations (as in Table 3).

### C.2 ACT TRAINING DETAILS

All ACT models were trained using 3 NVIDIA RTX A4000. To obtain a more thorough assessment of performance on ACT, we trained ACT policies for each task with new settings. We replace the visual backbone with DinoV2 (Oquab et al., 2023). Rather than padding episodes with the last frame, we modified the dataloader to randomly select an episode and a segment from the dataset.

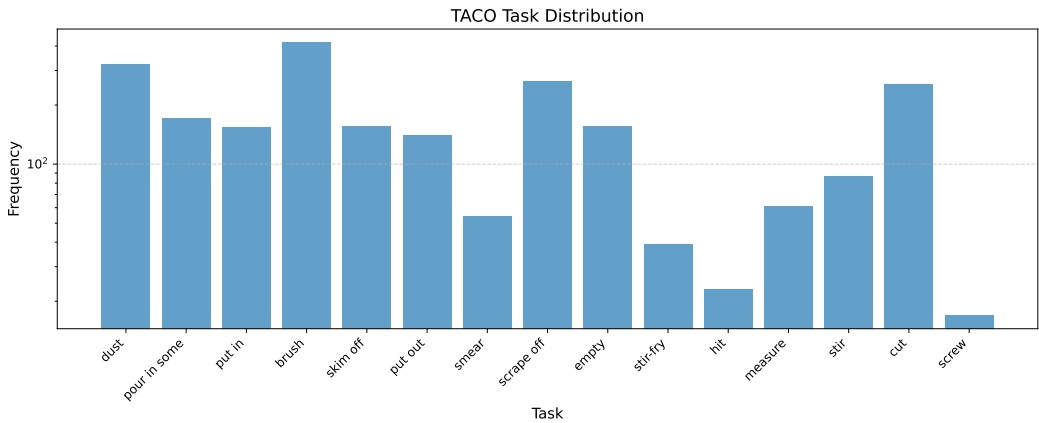

Figure 12: TACO (Liu et al., 2024) Task distribution

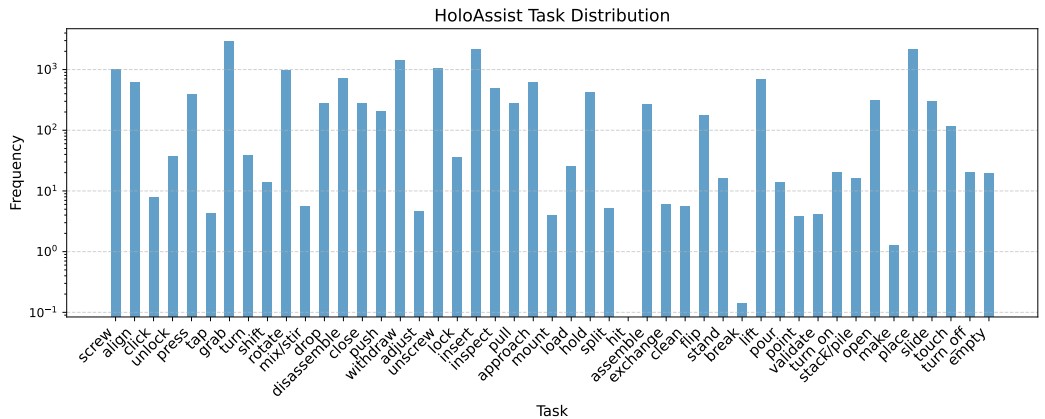

Figure 13: HoloAssist (Wang et al., 2023) task distribution

Table 3: Hyper Parameters for Pretraining on Human Videos

| Hyper Parameter Name | Pretraining On Human Video Data | Post-Training On Robot Demonstrations |
|---|---|---|
| Epoch | 20 | 115 |
| Batch Size | 16 * 8 * 4 | 16 * 8 * 4 |
| Learing Rate (LR) | 1e-4 | 2e-5 (2e-6 after 100epochs) |
| LR Scheduling | Cosine | Constant |
| Action Head Hidden Size | 1536 | 1536 |
| Action Head Transformer Layers | 6 | 6 |
| $\lambda_{\text{wrist trans}}$ | 20.0 | 20.0 |
| $\lambda_{\text{wrist rot}}$ | 5.0 | 5.0 |
| $\lambda_{\text{joint}}$ | 5.0 | 5.0 |

We set action chunk size to 64 for short horizon task Close-Drawer and 128 for the others. We use the following hyperparameters for training ACT models(as in Table 4).

Table 4: Hyper Parameters for ACT models

| Name | Value |
| --- | --- |
| Epoch | 50000 |
| Batch Size | 50 |
| Learing Rate | 1e-4 |
| Backbone | DinoV2 |
| Transformer Hidden Dimension | 512 |
| Encoder Layer | 4 |
| Decoder Layer | 7 |
| Number of Heads | 8 |

# D    DETAILED ACTION FORMULATION

## D.1    INFERENCE

As described in Section 3, our *EgoVLA* generates $H = 30$ actions per timestep at a frequency of 30 Hz, predicting actions for a 1-second future window. When deployed on the humanoid robot in simulation, the policy operates at 30 Hz. To enhance the smoothness of the robot's behavior and incorporate long-term planning capabilities, we employ action-chunking as described in ALOHA (Zhao et al., 2023b). For smoothing, we use a parameter value of 0.8.

## D.2    DETAILED MANO HAND PARAMETERIZATION

The MANO hand model (Romero et al., 2017) includes 15 ball joints with 45 degrees of freedom (DOFs). However, the human hand lacks this level of flexibility. To address this, MANO provides a low-rank representation using PCA. Consistent with HOI4D (Liu et al., 2022b) and HOT3D (Banerjee et al., 2024), we adopt the first 15 principal components of the PCA representation for hand pose parameterization, effectively representing the action space in a compact yet expressive form.

## D.3    RETARGETING DURING DEPLOYMENT

To ensure consistency between training and inference, we train a compact MLP for retargeting. Specifically, this retargeting MLP takes as input the 3D fingertip positions in the wrist frame of both hands and outputs the actuation values for all degrees of freedom (DOFs) of both hands.

The retargeting MLP is a four-layer neural network with hidden layer sizes of [64, 128, 64]. Its training data is derived from the process of converting robot demonstrations into human hand representations, as detailed in Sec.3. By employing this retargeting MLP, we not only achieve consistent retargeting results across training and inference but also enhance inference speed.

Table 5: Hyper Parameters for Retargeting MLP

| Name | Value |
|---|---|
| Epoch | 2000 |
| Batch Size | 2048 |
| Learing Rate | 0.001 |

# E HUMAN PREDICTION VISUALIZATION

In addition to the instruction-following results presented in Sec.5, this section provides additional visualizations of human trajectory predictions. We showcase the predicted future hand trajectories alongside the ground truth trajectories in Fig. 14. As illustrated, our *EgoVLA* produces accurate and consistent predictions across diverse scenes, object instances, and tasks. All these human videos are never seen by the model during training.

In Fig. 14a, the hand trajectory shows the approach to the knife, followed by lifting it and cutting the apple. Fig.14b and Fig.14c depict clear hand movements along the door's rotational axis during operation. In Fig. 14f, our model accurately predicts the placement position within the drawer. Lastly, in Fig.14d, the predictions remain consistent across different kettle instances, indicating the model's ability to understand the shared semantics of similar objects.

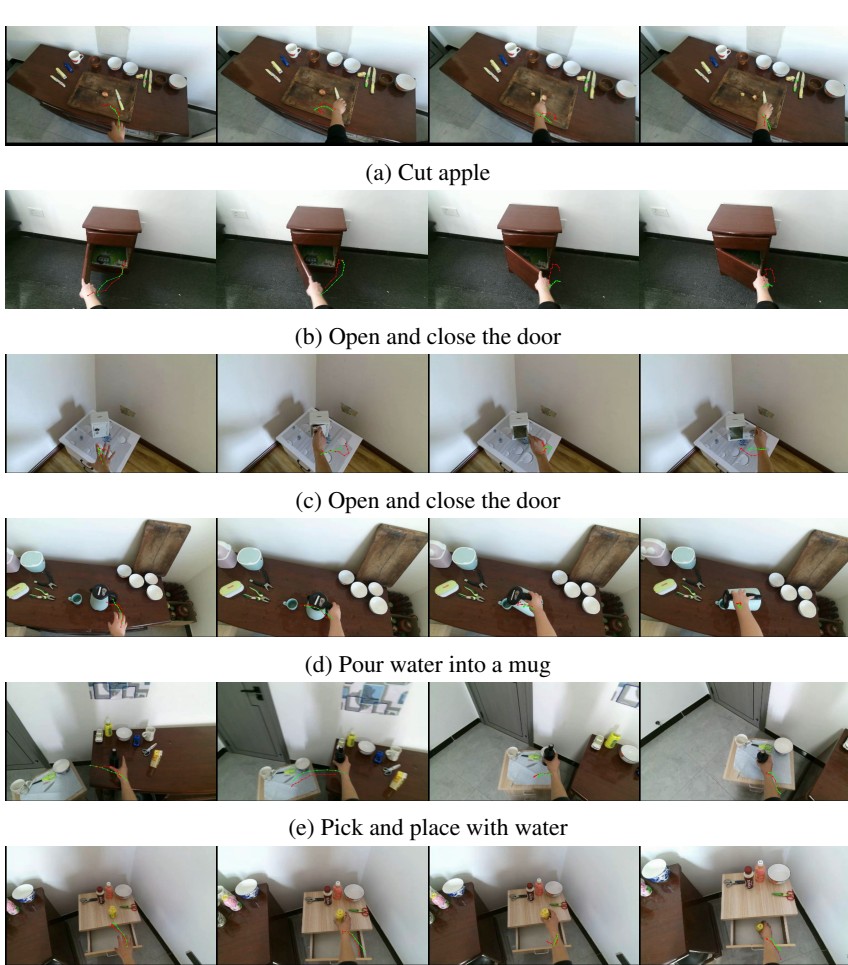

(a) Cut apple

(b) Open and close the door

(c) Open and close the door

(d) Pour water into a mug

(e) Pick and place with water

(f) Put it in the drawer

Figure 14: **Human Trajectory Prediction**: The green dotted lines are the future human hand trajectories predicted by our *EgoVLA* and the red dotted lines are the ground-truth future human hand trajectories. All sequences are from our evaluation set built upon HOI4D (Liu et al., 2022b) dataset. The language sequence descriptions are also provided as captions for each sequence.

## F    POLICY TRAJECTORIES VISUALIZATION

In addition to the policy trajectories discussed in Sec.5, this section provides additional visualizations of policy rollouts for various tasks. We present short-horizon task trajectories in Fig.15 and Fig.15. These figures highlight our model's ability to perform fundamental manipulation skills, including: Pushing: Illustrated in Fig. 15b. Grasping and Placing: Shown in Fig. 15d.

In Fig. 16, our model demonstrates its capacity to integrate these atomic skills from short-horizon tasks to solve diverse long-horizon tasks using a single generalist policy. For example: In Fig. 16d, the policy successfully executes a long-horizon task involving picking up a can, stacking the on the saucer and closing the drawer. In Fig. 16e, the policy achieves a sequence of actions to sort four different cans. This involves picking up each can and placing it in the correct container consecutively, demonstrating the model's accuracy and consistency in long-horizon tasks.

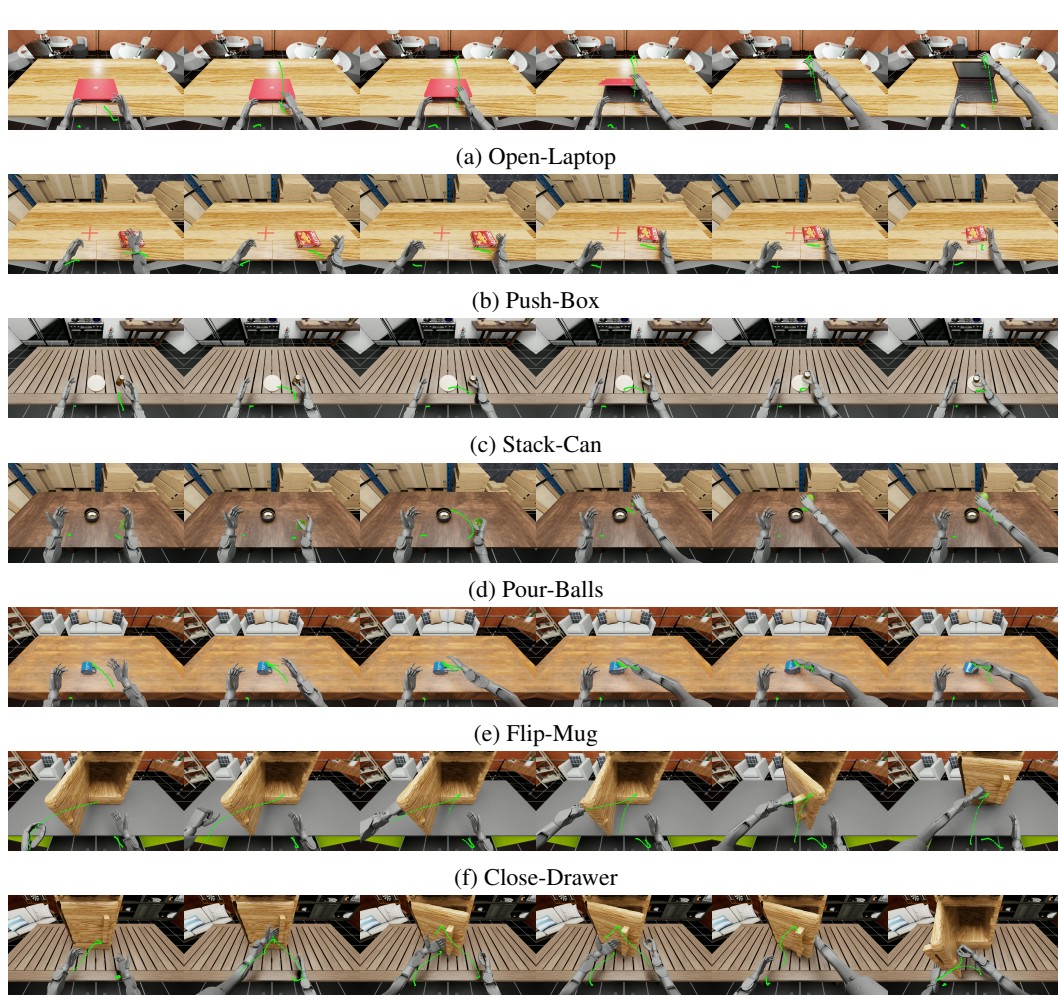

(a) Open-Laptop

(b) Push-Box

(c) Stack-Can

(d) Pour-Balls

(e) Flip-Mug

(f) Close-Drawer

(g) Open-Drawer

Figure 15: **Robot Trajectories for Short-Horizon Tasks**. We visualize the policy trajectories for short-horizon tasks in our *Ego Humanoid Manipulation Benchmark*. The green dotted lines are the future robot hand trajectories predicted by our *EgoVLA*.

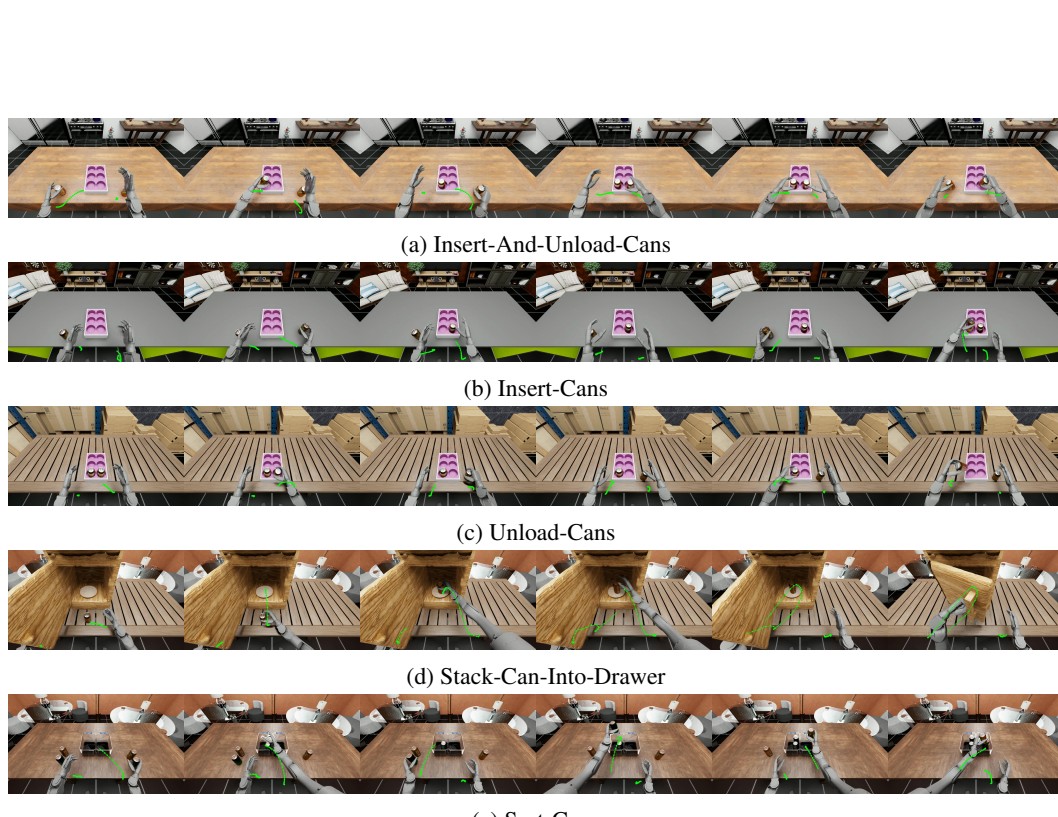

(a) Insert-And-Unload-Cans

(b) Insert-Cans

(c) Unload-Cans

(d) Stack-Can-Into-Drawer

(e) Sort-Cans

Figure 16: **Robot Trajectories for Long-Horizon Tasks.** We visualize the policy trajectories for long-horizon tasks in our *Ego Humanoid Manipulation Benchmark*. The green dotted lines are the future robot hand trajectories predicted by our *EgoVLA*.

## G  HUMANOID MANIPULATION BENCHMARK DETAILS

### G.1  ASSETS

The asset library combines models from NVIDIA Omniverse (NVIDIA, 2024b) and 3D-scanned meshes from HOI4D (Liu et al., 2022b), with modifications (e.g., added handles) to improve manipulability. We also fine-tune physics parameters to ensure realistic interactions while maintaining simulation efficiency.

### G.2  DETAILED TASK LISTS AND LANGUAGE INSTRUCTION FOR EVERY TASK

- **Pour-Balls**: "Pour balls in cup into bowl"
- **Push-Box**: "Push box to the marker"
- **Close-Drawer**: "Close the opened drawer"
- **Open-Drawer**: "Open the closed drawer"
- **Flip-Mug**: "Flip the mug"
- **Open-Laptop**: "Open the laptop"
- **Stack-Can**: "Put can on the saucer"
- **Sort-Cans**: "Put sprite cans to the left box, and orange cans to the right box"
- **Insert-Cans**: "Insert cans into the boxes"
- **Insert-And-Unload-Cans**: "Insert the right can into the slot and insert the left can into the slot, unload the right cans and then unload the left cans"
- **Unload-Cans**: "Unload the right cans and then unload the left cans"
- **Stack-Can-Into-Drawer**: "Put can on the saucer, and Close the drawer"

### G.3  OBJECT RANDOMIZATION RANGE VISUALIZATION

We provide the visualization for range of object randomization. Objects are randomized within a rectangle. The visualization of object at upper left and bottom right corners is shown in Fig. 17.

### G.4  ENVIRONMENT BACKGROUND VISUALIZATION

We provide a visualization of the diverse background of the environments. In total, we include 5 rooms and 5 tables, which can be combined in various configurations. The visualization of rooms and tables is shown in Fig. 18.

### G.5  TRAINING & EVALUATION VISUAL CONFIGURATION VISUALIZATION

We provide a visualization of the training and evaluation visual configuration. As shown in Fig. 19 and discussed in the main paper, we collect and train our models with three visual configurations and evaluate models on 22 unseen visual configurations.

### G.6  SIMULATION DETAILS AND SUCCESS & SUBTASK DEFINITIONS

To maintain a control frequency of 30 Hz, the simulation decimation and render interval are set based on the time step (dt), calculated as:

$$\text{simulation decimation} = \text{render interval} = \frac{1}{30 * dt}$$

For each environment, the detailed physics parameters and task definitions are provided below:

- **Push-Box**
  - Physics Parameters:
    * Physics simulation time-step in seconds ($dt$): $\frac{1}{120}$

(a) Close-Drawer: Drawer randomized within 0.2 x 0.1 (meters)

(b) Open-Drawer: Drawer randomized within 0.2 x 0.1

(c) Flip-Mug: Mug randomized within 0.2 x 0.2

(d) Pour-Balls: Bowl randomized within 0.2 x 0.2

(e) Insert-Cans: Cans randomized within 0.12 x 0.12

(f) Insert-And-Unload-Cans: Cans randomized within 0.12 x 0.12

(g) Unload-Cans: Cans and container randomized within 0.2 x 0.1

(h) Open-Laptop: Laptop randomized within 0.2 x 0.2

(i) Sort-Cans: Cans randomized within 0.12 x 0.12

(j) Push-Box: Box randomized within 0.2 x 0.2; Goal randomized within 0.06 x 0.2

(k) Stack-Can: Can and plate randomized within 0.2 x 0.2

(l) Stack-Can-Into-Drawer: Can randomized within 0.2 x 0.1; Plate and drawer fixed in vertical direction and randomized within 0.2 along horizontal axis

Figure 17: **Object Randomization Range**

* Box contact offset (in meters): 0.02
  - Task Definition:
    * Final task success if distance between box and goal marker is smaller than 0.08.
    * Subtask reach success if distance between end effector(EE) and box is smaller than 0.13.
- **Flip-Mug**
  - Physics Parameters:
    * dt: $\frac{1}{240}$
    * Mug collision approximation: SDF Mesh
    * Mug contact offset: 0.002

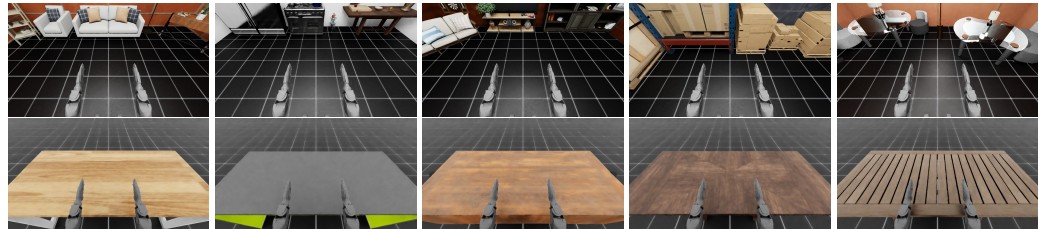

Figure 18: **Diverse Environment Background**: we included 5 different rooms and 5 different tables resulting 25 different visual configurations

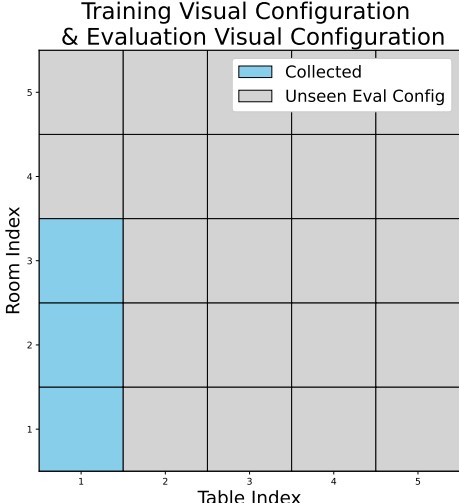

Figure 19: **Training & Evaluation Configuration**

- – Task Definition:
  - ∗ Final task success if z unit vector(unit vector in world frame pointing upwards) transformed by mug frame has z value greater than 0.5.
  - ∗ Subtask reach success if distance between EE and mug is smaller than 0.12.
- **Pour-Balls**
  - – Physics Parameters:
    - ∗ dt: $\frac{1}{240}$
    - ∗ Mug collision approximation: SDF Mesh
    - ∗ Mug contact offset: 0.002
    - ∗ Ball contact offset: 0.002
  - – Task Definition:
    - ∗ Final task success if at least 3 balls are within the mug
    - ∗ Subtask reach success if distance between EE and bottle is smaller than 0.12.
    - ∗ Subtask lift success if height of bottle increases by at least 0.06.
- **Sort-Cans**
  - – Physics Parameters:
    - ∗ dt: $\frac{1}{120}$
    - ∗ Can collision approximation: Convex hull
    - ∗ Mug contact offset: 0.0015
    - ∗ Container collision approximation: Convex Decomposition
    - ∗ Container contact offset: 0.0015
  - – Task Definition:

* Final task success if both Sprite cans are inside the left container and both Fanta cans are inside the right container
* Subtask reaches success if the distance between EE and can is smaller than 0.12. Value of subtask reach represents the number of cans that satisfy the condition.
* Subtask lift success if height of the can increases by at least 0.06. Value of the subtask lift represents the number of cans that satisfy the condition.
* Subtask sort represents the number of cans that are inside the correct container.

- **Insert-Cans**
  - Physics Parameters:
    * dt: $\frac{1}{120}$
    * Can collision approximation: Convex hull
    * Can contact offset: 0.0015
    * Container collision approximation: SDF Mesh
    * Container contact offset: 0.002
  - Task Definition:
    * Final task success if cans x,y coordinates are within the container and the z value is smaller than the threshold for inserted cans (1.065)
    * Subtask reaches success if the distance between EE and can is smaller than 0.12. Value of subtask reach represents the number of cans that satisfy the condition.
    * Subtask lift success if height of the can increases by at least 0.06. Value of the subtask lift represents the number of cans that satisfy the condition.
    * Subtask insert represents the number of cans that are inserted.

- **Unload-Cans**
  - Physics Parameters:
    * dt: $\frac{1}{120}$
    * Can collision approximation: Convex hull
    * Can contact offset: 0.0015
    * Container collision approximation: SDF Mesh
    * Container contact offset: 0.002
  - Task Definition:
    * Final task success if cans x,y coordinates are not within the container, and can is standing upwards on the table (same calculation as Flip-Mug), and height of can is greater than table surface
    * Subtask reach success if distance between EE and can is smaller than 0.12. Value of subtask reach represents the number of cans that satisfy the condition.
    * Subtask lift success if height of can increases by at least 0.06. Value of subtask lift represents the number of cans that satisfy the condition.
    * Subtask unload represents the number of cans that are unloaded.

- **Insert-And-Unload-Cans**
  - Physics Parameters:
    * dt: $\frac{1}{120}$
    * Can collision approximation: Convex hull
    * Can contact offset: 0.0015
    * Container collision approximation: SDF Mesh
    * Container contact offset: 0.002
  - Task Definition:
    * Final task success if cans are inserted first then unloaded
    * Subtask reach success if distance between EE and can is smaller than 0.12. Value of subtask reach represents the number of cans that satisfy the condition.
    * Subtask lift success if height of can increases by at least 0.06. Value of subtask lift represents the number of cans that satisfy the condition.
    * Subtask insert represents the number of cans that are inserted.
    * Subtask unload represents the number of cans that are unloaded.

- **Close-Drawer**
  - Physics Parameters:
    * dt: $\frac{1}{240}$
    * Drawer body collision approximation: Convex Decomposition
    * Drawer door collision approximation: SDF Mesh
    * Drawer contact offset: 0.01
  - Task Definition:
    * Final task success if drawer is fully closed
    * Subtask reach success if distance between EE and handle on drawer door is smaller than 0.12
    * Subtask move-door success if drawer is closed at least 10% of its joint range

- **Open-Drawer**
  - Physics Parameters:
    * dt: $\frac{1}{240}$
    * Drawer body collision approximation: Convex Decomposition
    * Drawer door collision approximation: SDF Mesh
    * Drawer contact offset: 0.01
  - Task Definition:
    * Final task success if drawer is fully open
    * Subtask reach success if distance between EE and handle on drawer door is smaller than 0.12
    * Subtask move-door success if drawer is open at least 10% of its joint range

- **Stack-Can**
  - Physics Parameters:
    * dt: $\frac{1}{120}$
    * Can collision approximation: Convex hull
    * Can contact offset: 0.0015
    * Plate collision approximation: Convex Decomposition
    * Plate contact offset: auto computed
  - Task Definition:
    * Final task success if horizontal distance (calculated using x,y coordinates) between can and plate is smaller than radius of plate and difference in z values is smaller than 0.02
    * Subtask place success if horizontal distance (calculated using x,y coordinates) between can and plate is smaller than radius of plate and difference in z values is smaller than 0.05

- **Stack-Can-Into-Drawer**
  - Physics Parameters:
    * dt: $\frac{1}{240}$
    * Can collision approximation: Convex hull
    * Can contact offset: 0.0015
    * Plate collision approximation: Convex Decomposition
    * Plate contact offset: auto computed
    * Drawer body collision approximation: Convex Decomposition
    * Drawer door collision approximation: SDF Mesh
    * Drawer contact offset: 0.01
    *
  - Task Definition:
    * Final task success if horizontal distance (calculated using x,y coordinates) between can and plate is smaller than radius of plate, difference in z values is smaller than 0.02, and drawer is close.
    * Subtask reach_drawer success if distance between left EE and drawer handle is smaller than 0.12.

* Subtask reach_can success if distance between right EE and can is smaller than 0.12.
* Subtask lift success if height of can increases by at least 0.06.

- **Open-Laptop**
    - Physics Parameters:
        * dt: $\frac{1}{120}$
        * Laptop body collision approximation: Convex Hull
        * Laptop lid collision approximation: Convex Hull
        * Laptop lid contact offset: auto computed
    - Task Definition:
        * Final task success if laptop is open by at least 70% of its joint range.
        * Subtask move_lid_success if laptop is open by at least 15% of its joint range.

For robot hands, collision approximation for each link is Convex Decomposition, and contact offset is 0.005.

