# OpenReview forum: "EgoVLA: Learning Vision-Language-Action Models from Egocentric Human Videos"
_ICLR.cc/2026/Conference — Submitted to ICLR 2026_

### Official Review · Reviewer_9rjm · 2025-10-29

**Soundness:** 3
**Presentation:** 3
**Contribution:** 2
**Rating:** 4
**Confidence:** 4

**Summary:**

This paper presents EgoVLA, a new vision-language-action (VLA) model that is trained on large-scale egocentric human video data. The authors have preprocessed the existing egocentric human data to get MANO annotations. Then, EgoVLA is trained to predict the ground-truth MANO parameters, including wrist poses and hand joint angles. Finally, EgoVLA is fine-tuned on a small amount of robot demonstration data. This process involves 1) converting the robot data to human representations for training and 2) mapping from human hand to robot hand in the inference stage. Furthermore, the authors have proposed a new benchmark for evaluating EgoVLA, which is based on NVIDIA Issac Lab. Experiments show that pretraining VLAs on egocentric video data improves task success and generalization.

**Strengths:**

- [S1] The paper explores a novel direction how we can train VLA models using egocentric human video data.
- [S2] The paper proposes a new benchmark for evaluating VLA models, especially based on humanoid robots.
- [S3] The paper is well-written and easy to understand.

**Weaknesses:**

- [W1] As the authors note in their limitations section, egocentric video data still requires hand and wrist pose annotations, which are difficult to collect. This suggests that egocentric video data faces similar data scalability challenges as robot data collection.
- [W2] In the experiments, the authors only compare EgoVLA against a simple non-VLA baseline—ACT [1]—making it difficult to assess the true performance gains and whether the proposed method outperforms other strong alternatives, such as π0 [2] and OpenVLA [3]. Benchmarking against such competitive models would strengthen both the validation of the proposed method and the credibility of the benchmark itself.



**References**
- [1] Zhao et al., Learning fine-grained bimanual manipulation with low-cost hardware, 2023.
- [2] Black et al., $\pi_0 $: A Vision-Language-Action Flow Model for General Robot Control, 2024.
- [3] Kim et al., Openvla: An open-source vision-language-action model, 2024.

**Questions:**

N/A

---

### Official Review · Reviewer_4urF · 2025-10-30

**Soundness:** 3
**Presentation:** 3
**Contribution:** 2
**Rating:** 4
**Confidence:** 3

**Summary:**

The paper addresses the problem of learning vision-language-action (VLA) models from egocentric human videos. The proposed model predicts human wrist and hand poses, which are subsequently retargeted to robot actions using inverse kinematics. A small number of robot demonstrations are employed for model fine-tuning. To assess the framework’s effectiveness, the authors introduce a simulation benchmark for systematic evaluation.

**Strengths:**

Leveraging human videos to learn dexterous manipulation policies is a promising direction, particularly given the high cost and effort required to collect teleoperation data.

Aligning the action space between humans and robots through rigid 3D transformations and retargeting is a sound and well-motivated design choice.

Overall, the proposed method demonstrates strong performance based on the reported evaluations.

**Weaknesses:**

The full system is evaluated only in simulation, and the domain gap between simulation and the real world is not adequately addressed.

Moreover, the method still requires task-specific robot data for fine-tuning, which limits its scalability and generalization potential.

**Questions:**

Human motions are highly versatile, and some may be infeasible to transfer to robotic embodiments through inverse kinematics or retargeting due to morphological or dynamic mismatches. How do the authors plan to handle such cases?

What are the common failure modes of the proposed method? A detailed analysis of these limitations would help readers better understand the strengths and weaknesses of the approach.

Have the authors explored zero-shot deployment of the system---i.e., using only human data without robot fine-tuning---to assess its generalization capability?

---

### Official Review · Reviewer_v3pr · 2025-11-01

**Soundness:** 2
**Presentation:** 2
**Contribution:** 2
**Rating:** 2
**Confidence:** 4

**Summary:**

The manuscript proposes a VLA model capable of learning bimanual robot manipulation from egocentric human demonstration videos. This is performed by defining a common representation consisting of wrist pose and hand pose.

To this end, the authors build a dataset consisting of four publicly available subsets and perform preprocessing making the data suitable for training. This consists of harmonizing data, performing camera motion compensation, adding language placeholders, and providing hand pose in the MANO format.

In a first stage, the VLA model is trained on human data, predicting the future sequence of wrist pose and hand pose. In a finetuning stage, the model is trained on a small robot dataset, predicting handposes that are retargeted to the robot pose. For deployment on a humanoid robot, hand poses are converted to robot joint parameters using an MLP.

The model is evaluated on the proposed Ego Humanoid Manipulation Benchmark.

**Strengths:**

- The model allows joint learning from robotic and non-robotic data, approaching a core problem of VLA training
- The proposed VLA is well engineered matching human hand representations with dexterous robot manipulators at each training stage
- The model shows improved results over baseline methods on the proposed Ego Humanoid Manipulation Benchmark

**Weaknesses:**

- The method only utilizes simple human demonstrations with clearly visible hands in simple environments. No results on day-to-day demonstrations are presented, limiting the potential advantage of reduced dataset collection cost. Showing generalization to these scenarios could potentially be done on datasets like Ego4D or Epic Kitchens.
- EgoVLA is purely evaluated on simulation results. With VLA models often showing considerably different performance on real-word deployment, this shows little evidence of the model's actual performance.
- The proposed Ego Humanoid Manipulation Benchmark is not validated to correlate with real-world robot performance. While a direct match between simulation and real-world performance is often not given, the benchmark should be validated to ensure method rank consistency.

**Questions:**

- The visualization show well detected hand poses. It would be interesting to understand how sensitive model training is to incorrect detections in real-world scenarios.

---

### Official Review · Reviewer_heQB · 2025-11-02

**Soundness:** 3
**Presentation:** 3
**Contribution:** 2
**Rating:** 4
**Confidence:** 4

**Summary:**

This paper explores whether we can train manipulation policies without relying on large amounts of robot data.  The authors argue that robot action spaces can be close enough. Human hand and wrist motions can be converted into robot actions using simple geometric transforms like inverse kinematics and hand retargeting. The authors train a VLA model purely on egocentric human videos and then do a small amount of robot fine-tuning, resulting in a robot-ready policy called EgoVLA. They also introduce a new bimanual manipulation benchmark in Isaac Sim with diverse tasks and demonstrations.

**Strengths:**

* The paper addressed an interesting and important problem.

* The method proposed in this paper is intuitive, and the paper is easy to follow.

* The proposed benchmark, although limited, is valuable. Within this benchmark, the proposed method outperforms both specialist and generalist baselines and generalizes better across viewpoints and object positions.

**Weaknesses:**

* The main limitation is that validation is mostly in simulation. While the contributions may still be meaningful for scaling robot learning with accessible human video data, the current justification based on only the simulation is not convincing. This is particularly important because the argument made in this paper is bold, which can also be justified through real-world experiments. In other words, the made claims may be true only within this paper's experimental settings.

* While being intuitive and desirable, it is hard to admit the closeness of the robot and human action spaces. This is primarily due to the make of the human hands being unlike of robots, therefore the offered degree of freedoms and the articulation, flexibility, and dexterity of human hands. Certain actions may simply be infeasible for robots that are very simple for human hands. In some constrained setting, it may, however, be true what the paper argues. However, that line is not made clear; therefore, such a bold claim is not to be taken lightly.
This claim is particularly worrisome, as it turns out to be empirical  (besides being intuitive, thus possibly deceptive) and the experimental claims are made through the self-made simulation benchmark.

**Questions:**

* Please delineate the claim similarity of the action spaces of humans and robots in more formal terms, while acknowledging the fact that the current claim is not true in general settings with current robots (or the one used in the experiments).

* The main limitation is the real robot experimental results. This is particularly important for the argument put forward in this paper. Without those experiments, the paper cannot be accepted at its current stage, in addition to the requested delination.

* Please provide the details on how the ego-motion is compensated to bring the hand pose in the first view reference frame. Are the ground truth camera poses available at all times?

* Ablation study: while using the wrist pose, is the location of the wrist sufficient to achieve the final results? Do the other pose variables (rotation) play a vital role as well? Is the wrist pose and end effector assumed to be aligned? If so,  how easy is such alignment?

---

### Meta-Review · Area_Chair_1ukx · 2026-01-05

**Summary:**

The reviewers generally found the problem setting and overall direction promising, as leveraging large-scale egocentric human videos for learning manipulation policies could significantly reduce the cost of robot data collection.

However, several concerns were raised regarding the conceptual framing and validity of the proposed approach. In particular, reviewers questioned the assumption that human and robot action spaces can be meaningfully aligned, the interpretation of the claimed “joint action space”, and whether the proposed model truly constitutes a Vision-Language-Action (VLA) model. Additional concerns relate to potential biases introduced by human egocentric motion and the reliance on simulation-only evaluations. These concerns collectively informed the final recommendation.

**Reviewer Concerns:**

Concerns addressed by the rebuttal:
None. The paper did not receive a substantive rebuttal addressing the reviewers’ conceptual and methodological concerns.

Existing concerns:
First, while the paper claims to construct a “joint action space” between humans and robots, the proposed space is not truly joint in the sense of a shared latent action representation, but instead relies on hand-engineered retargeting and inverse kinematics. As a result, the claim around a joint action space (e.g., Figure 4) may be misleading.

Second, the alignment between human and robot hand poses remains only approximate. Human hand trajectories and kinematics do not perfectly correspond to robotic embodiments, and a layer of retargeting is still required.

Third, there is a potential bias between egocentric human data and robot manipulation. Human hand trajectory prediction inherently entangles hand motion with ego-motion (e.g., head or body movement), whereas robot platforms often operate with largely fixed
or limited viewpoints. The paper does not sufficiently analyze how this mismatch affects the effectiveness of human data augmentation or the generalization of the learned policies.

Finally, the experiments are evaluated in simulation and the paper lacks real robot experiments.

**Reviewer Scores:**

Reviewer heQB: Likely to maintain their original score, as their concerns regarding action space alignment and conceptual framing were not addressed.

Reviewer v3pr: Unlikely to change their score. The absence of a rebuttal leaves the key concerns about novelty, evaluation scope, and generalization unresolved.

Reviewer 4urF: Likely to maintain their score. While they were positive about the direction, the lack of clarification on the core conceptual issues limits stronger support.

Reviewer 9rjm: Unlikely to change their score, given that concerns about bias between human egocentric data and robot manipulation remain unaddressed.

---

### Decision · Program_Chairs · 2026-01-26

Reject